# The Mistletoe and Breast Cancer (MAB) Study: A UK Mixed-Phase, Pilot, Placebo-Controlled, Double-Blind, Randomised Controlled Trial

**DOI:** 10.3390/cancers17193169

**Published:** 2025-09-29

**Authors:** Lorna J. Duncan, Susan Bryant, Gene Feder, Maria Gresham, Poppy Gibson, Debbie Sharp, Jeremy P. Braybrooke, Alyson L. Huntley

**Affiliations:** 1Centre for Academic Primary Care, University of Bristol, Bristol BS8 2PS, UK; lorna.duncan@bristol.ac.uk (L.J.D.); susan.bryant@bristol.ac.uk (S.B.); gene.feder@bristol.ac.uk (G.F.); debbie.sharp@bristol.ac.uk (D.S.); 2Bristol Medical School, University of Bristol, Bristol BS8 1UD, UK; maria.hancock.2018@bristol.ac.uk (M.G.); uk19574@bristol.ac.uk (P.G.); jeremy.braybrooke@uhbw.nhs.uk (J.P.B.); 3Bristol Haematology and Oncology Centre, University Hospitals Bristol and Weston NHS Foundation Trust, University of Bristol, Bristol BS1 3NU, UK

**Keywords:** *Viscum album*, mistletoe, breast cancer, feasibility, pilot randomised controlled trial, quality of life

## Abstract

**Simple Summary:**

The Mistletoe and Breast Cancer (MAB) randomised placebo-controlled, double-blind trial investigates the feasibility of the use of the herbal medicine mistletoe (*Viscum album*) as an adjunct supportive therapy alongside conventional cancer treatment within the National Health Service in the United Kingdom.

**Abstract:**

Background/Objective: To test the feasibility of a mixed-phase, pilot, placebo-controlled, double-blind trial of mistletoe therapy (MT) with an embedded qualitative study in the UK National Health Service (NHS) setting. Methods: The aim was to recruit 45 patients via an NHS oncology centre with a diagnosis of early or locally advanced breast cancer. Participants were allocated to Iscador^®^ Malus, Iscador^®^ Pinus, or physiological saline (placebo). Diaries and quality-of-life questionnaires were administered. Qualitative interviews were conducted with participants, oncologists, and nurses. Feasibility was assessed by recruitment, retention, adherence, blinding, and safety. Results: Sixty-seven patients were approached between August 2019 and March 2020, 15 gave consent, 14 participants were randomised, and 2 withdrew during the trial. Ten participants and five staff were interviewed. Barriers to recruitment were the additional treatments/time, extra injections, and the possibility of placebo allocation. Adherence was very good whilst the participants were on the study therapy. Diaries and interviews indicated that 11/14 participants struggled with injections and skin reactions. There were 22 adverse events due to the MT, related to the injections or skin reactions. Conclusion: This pilot study examined the feasibility of conducting a randomised placebo-controlled, double-blind trial of mistletoe therapy for breast cancer patients within the UK NHS. The results describe the challenges and achievements of recruitment, retention, adherence, blinding, and safety in this context.

## 1. Introduction

Anthroposophic use of mistletoe has developed using specific mistletoe formulations and regimens with a focus on supportive cancer care [1]. In Europe, mistletoe is a commonly used therapy for patients with cancer, and it is integrated into oncology care via health insurance schemes in Germany and Switzerland [1]. Neither the European Medicines Agency (EMA) nor the Food and Drug Administration (FDA) states any opinions or recommendations on mistletoe therapy (MT), although they have both compiled summaries of MT evidence [2,3].

A recent Health Technology Assessment (HTA) report that focused on MT for breast cancer patients concluded that improvements in health-related quality of life compared to the control group were small to moderate, but that the risk of bias within efficacy trials was high [4]. This evidence is based on seven clinical trials (three were randomised controlled trials (RCTs)) using mistletoe as an adjunct therapy to the standard treatment of chemotherapy and or radiotherapy in patients with breast cancer. Local skin reactions of low and moderate severity were reported in a median of 25% (range, 5 to 94%) of patients, and mild to moderate systemic reactions in a median of 2% (range, 0 to 8%) of patients.

Observational studies, cross-sectional studies, and an ethical evaluation were also included in this HTA report. A comparative cost analysis from Germany reported significantly lower medical costs within 5 years after surgery for patients with MT than for patients without it; however, there was no control for systematic bias. From studies conducted in Germany, it was calculated that a median of 25% (range, 7–46) of patients with breast cancer and 29% of treatment providers use MT. The main motivations to use MT cited by patients in these studies were to reduce side effects, strengthen the immune system, and take an active role in the treatment process. It was also reported in these studies that patients felt insufficiently advised about MT.

Whilst MT is prescribed in the UK, this is through the private sector, and predominantly in primary or community care [5]. The lack of a robust, relevant evidence base for MT in supportive cancer care prohibits progress in this field, despite its potential to improve the patient experience of cancer care, a major priority of the National Health Service (NHS) cancer plan [6]. The NHS does not specifically give information or recommendations on MT, but it does advise caution to cancer patients considering herbal therapy during their treatment [7].

As MT is not available within the NHS, the UK population is generally unaware of MT and its application as a supportive adjunct therapy for cancer. This potentially enhances participant blinding to treatment allocation in a randomised clinical trial in the UK, as they are less likely to understand MT’s appearance and its effects.

Our aim was to test the feasibility of a mixed-phase, pilot, placebo-controlled, double-blind RCT of MT in patients with breast cancer in the UK NHS setting.

## 2. Materials and Methods

### 2.1. Design and Setting

We conducted a pilot, placebo-controlled, double-blind RCT of MT with an embedded qualitative study in patients with breast cancer in a UK NHS setting [8]. There were three groups in the trial: Iscador Malus (M), Iscador Pinus (P) MT, and physiological saline as the placebo. The feasibility of the trial was assessed in terms of recruitment, retention, adherence, blinding, and safety (EudraCT number: 2018-000279-34). The MAB study adheres to the CONSORT guidelines [9].

### 2.2. Sample Size and Site

The aim was to recruit 45 adult patients via one site, the Bristol Haematology and Oncology Centre (BHOC) at University Hospitals Bristol and Weston NHS Foundation Trust. This was a feasibility study with the sample size chosen to allow for fair assessment of the aims of recruitment, retention, and completion of outcomes and an assessment of the viability of blinding.

### 2.3. Inclusion and Exclusion Criteria

Participants were adults 18 years or over with histologically confirmed early or locally advanced invasive breast cancer who, following surgery, were to receive adjuvant chemotherapy, with or without radiotherapy, and were able to be randomised within 12 weeks of surgery. Patients could be included if they were also scheduled to receive biological therapies such as trastuzumab or endocrine therapy. Patients who received only radiotherapy were excluded, as this treatment is generally well tolerated and of short duration. They had to be willing to self-administer or have a nominated person administer their injections. Their Eastern Cooperative Oncology Group (ECOG) performance status was 0 or 1, and they could have no active, uncontrolled infection [10]. Full inclusion and exclusion criteria are available in the Appendix A.

### 2.4. Randomisation and Blinding

The patient randomisation list and the medication block randomisation lists were produced by an in-house statistician at Iscador AG. Randomisation of patients was conducted by University Hospitals Bristol Pharmacy (UHBP). Allocation of participants to Iscador^®^ M/Iscador^®^ P/placebo in a 1:1:1 ratio was performed by UHBP. A separate randomisation list was held by UHBP for individual emergency unblinding. In the case of a serious adverse event and unblinding being required, the pharmacist would have been asked by the principal investigator to look at the unblinding randomisation list. Both participants and healthcare professionals were uninformed about the allocation of the trial therapies. Details of the randomisation process are available in the protocol paper [8].

Participants took the MAB study therapy for approximately six months. The first study therapy was given within a week of randomisation to the study and, ideally, prior to the start of chemotherapy. If participants had a skin reaction larger than 5 cm in diameter, they stopped the MAB study therapy for one week and then resumed with a step-down in dose. Participants or participant-nominated individuals (e.g., spouse or partner) were taught to administer the MAB study therapy by the research nurse—once by nurse demonstration and then by administering the therapy themselves under nurse observation.

### 2.5. Mistletoe Therapy (Intervention Group)

Participants received mistletoe preparations Iscador^®^ M or Iscador^®^ P in 1 mL ampoules for subcutaneous injection. The standard therapy regimen was devised from the manufacturer’s recommendation in conjunction with the MAB advisory group (Appendix A).

### 2.6. Placebo (Control Group)

Placebo participants underwent the same regimen as those in the intervention group, and the placebo ampoules had identical external packaging and labelling to the MT ampoules. The placebo was physiological saline, 0.90% *w*/*v* of sodium chloride, (308 mOsm/L) (Appendix A).

### 2.7. Data Collection

Feasibility was measured using clinical study data and qualitative interview data to assess the following objectives:

#### 2.7.1. Recruitment

Recruitment rate.Obstacles to recruitment.

#### 2.7.2. Retention and Adherence

Attrition rate with reasons, if possible.Acceptability of regular subcutaneous injections.Adherence to the study therapy schedule.Assessment of therapy-related symptoms and health-related quality of life in the sample population.Completion of outcome measures.

#### 2.7.3. Blinding

Assessment of blinding of patients.

#### 2.7.4. Adverse Events

Adverse events from MT and placebo subcutaneous injections.

### 2.8. Clinical Study Data

#### 2.8.1. Participant Diaries

Participants received a diary card pack to record their thrice-weekly MAB study therapy administration, the reaction to the injection, and any comments they wanted to add (Appendix A).

#### 2.8.2. Questionnaire Pack

This comprised six questionnaires: European Organisation for Research and Treatment of Cancer Quality of Life (EORTC-QLQ-C30), European Organisation for Research and Treatment of Cancer Quality of Life—Breast Cancer (EORTC QLQ-BR23), Functional Assessment of Cancer Therapy—Neutropenia (FACT-N) scale, Cancer Fatigue Scale, Autonomic Regulation Scale—State (ARS-S), and CompleMentary and Alternative Beliefs Inventory (CAMBI) [11,12,13,14,15,16]. These were administered at three time points: time point zero or baseline (T0)—following randomisation and before the start of the chemotherapy regimen; time point one (T1)—following the 3rd cycle of chemotherapy; and time point two (T2)—four weeks after last standard treatment (chemotherapy with or without radiotherapy), on the day of the last study (MAB study therapy) treatment.

T0 questionnaires were completed during an initial visit to the BHOC. T1 and T2 were either generated by the REDCAP database (for four participants) or administered by BHOC staff at the appropriate time for those participants working on this paper.

#### 2.8.3. Adverse Events

All adverse events experienced by the participants during the MAB trial were collected independent of their causal relationship with MT. The manual of the Common Terminology Criteria for Adverse Events (CTCAE) grading of toxicity was used to classify the adverse events [17].

#### 2.8.4. Qualitative Interviews

Semi-structured interviews were conducted by LD with participating patients, oncologists, and research nurses. LD is an experienced qualitative researcher based at the University of Bristol with a research interest in complementary therapies. Participating patients were initially approached by the research nurses. If they agreed, the research nurses passed their contact details to LD, who then sought consent and arranged an interview. LD approached BHOC staff directly. All patient participants who had consented to being interviewed were contacted.

Initial interviews in the first few weeks of the MAB study therapy were conducted in person with five participants between October 2019 and February 2020. These five interviews were conducted in a place agreed upon with the participants: Canynge Hall (University of Bristol) (2), quiet spots within public cafes in Bristol (2), and in a participant’s workplace (1). All remaining interviews were undertaken remotely in compliance with COVID-19 pandemic restrictions, and interviewees selected either telephone or online options. Thus, all staff interviews, five initial participant interviews, and all ten second interviews, which took place up to one month after all treatment was completed, were conducted remotely. Written consent was obtained, and topic guides were used to aid questioning whilst allowing interviewees to discuss additional issues (Appendix A). Topics considered included (i) awareness of MT; (ii) recruitment; (iii) retention and adherence; (iv) blinding; (v) perspectives on complementary and alternative therapies; and (vi) availability of complementary and alternative therapies within the NHS (staff only).

Interviews were transcribed and coded in NVivo [version 12.0] using themes broadly linked to those used in the interview topic guides (Appendix A). A qualitative inductive analysis approach was used, guided by the Braun and Clarke methods [18]. Initial coding and development of themes were performed by LD and AH, with the remaining coding and synthesis performed by LD, MG, and PG. Narrative synthesis of the themes from the perspectives of participants and staff is reported using the interview data, as well as relevant items from participant diaries and questionnaires [19].

### 2.9. Data Analysis

The recruitment rate is expressed using descriptive statistics, as are all questionnaires completed by the participants. Retention is summarised by recording the number of participants in each study group at the pre-specified worst toxicity time point (T1) and 4 weeks post-standard treatment (T2). A questionnaire was considered completed if data could be used. For example, the EORTC QLQ-C30 data was included if at least half the questions from the factors of interest were complete in accordance with the manual of the EORTC Quality of Life group. Blinding was assessed by asking the patient at T2 to assess which study treatment they had received or to register a “don’t know” option.

### 2.10. MAB Management

The core MAB management team members were SB, LD, AH, and GF, working in conjunction with the BHOC team led by JB. In the development and approval process of the MAB study, this core team worked with a wider team of co-applicants, an advisory group, and BHOC colleagues detailed within the acknowledgement section. The core MAB team met monthly during the MAB trial, with additional attendance by patient and public involvement (PPI) members and advisory members as required.

The MAB Trial Steering, Data Monitoring, and Ethics Committee comprised clinical, statistical, and patient representation (see Acknowledgement section). The committee met regularly (every 4–5 months) prior to trial start-up until the end, remaining available beyond that time as needed.

### 2.11. Patient and Public Involvement

Bristol-based MT research was formally launched in September 2010 with a public education event led by three members of the MAB team and the Cancer Research Network (CRN) of West of England cancer lead, Catherine Carpenter Clawson, and involved the public, patients, and health professionals. This event contributed to our protocol development and the initial development of our topic guide for our qualitative interviews, with two members of the MAB team piloting questions with health professionals. The MAB PPI group was convened in September 2017 and comprised four women with the experience of a breast cancer diagnosis, treatment, and beyond. This initial work contributed to the refinement of our patient interview topic guide and patient-facing documents and to the use of a transparent disc to aid measurement of anticipated skin reactions during the trial. Two of our PPI members remained actively involved throughout the trial, joining team meetings and the trial steering group meetings.

## 3. Results

### 3.1. Recruitment

Sixty-seven potential participants were approached; screened (twenty-five were not eligible); and, if appropriate, recruited by the research nurses between 1 August 2019, and 19 March 2020. Following screening, 15 were eligible and gave their consent. Participant 15 dropped out prior to randomisation.

Characteristics are available for 14 of the participants. All were women and had a mean age of 49 (range 36–76) years. They identified as White British (*n* = 11), White other background (*n* = 2), and Chinese (*n* = 1). (Breast cancer statuswas stage 1–3, and 10/14 participants were Estrogen Receptor Positive (ER +) (Table 1).

Ten of the fourteen participants agreed to be interviewed both at the beginning and end of the study (20 interviews). The mean age of those interviewed was 46 years (range, 36–63); eight participants described themselves as White British and the remaining two as White other background. Five interviews were also undertaken with staff—two consultant oncologists involved in recruitment to the study and three research nurses who were the day-to-day points of contact for the MAB study participants.

Qualitative analysis of the interviews was based on the feasibility outcome framework (A–F) with illustrative quotes (Table 2). (A) Recruitment: awareness of mistletoe therapy, barriers to recruitment, and enablers to recruitment. (B) Retention and adherence: barriers to retention and adherence; enablers to retention and adherence. (C) Assessment of blinding. (D) Adverse events. (E) Questionnaire data. (F) Complementary and alternative medicine beliefs.

All patient participants and staff interviewees were asked what they knew about MT before learning of the trial. The unanimous response from participants in the qualitative interviews was that they had not heard of the therapy before, although everyone was familiar with the mistletoe plant; all were surprised to hear of its therapeutic use. (Table 2, A1 and A2).

The nurses were also unfamiliar with MT, although one said patients often spoke of complementary therapies with her and that mistletoe might have been mentioned. One of the oncologists was aware of patients who had used MT in private practice. (Table 2, A3).

#### 3.1.1. Barriers to Recruitment

In the first five months of recruitment (from August 2019 to December 2019), 43 potential participants were approached, and 6 (14%) gave their consent. From January 1st to March 19th, 2020, 24 potential participants were approached, and 9 (38%) gave their consent. The difference in these recruitment rates was related to an initial ambiguity in the inclusion criteria around the human epidermal growth factor receptor 2 (HER2) status of patients, for which a major protocol amendment was made, improving recruitment from January 2020 onwards.

Reasons for non-participation can be grouped into four categories: not meeting the screening criteria (25), patient decision (22), healthcare professional decision (1), and practical barriers (4). (Figure 1) By 19 March 2020, 15 participants were consented, and 14 were randomised to treatment allocation. The 15th participant withdrew before randomisation, citing the COVID-19 pandemic as the reason. The most common reasons recorded in the recruitment log for women not accepting the invitation into the trial (82%) were the injections or the related additional visits to the BHOC. These concerns were echoed in the qualitative interviews, with some participants expressing reservations around self-injections and extra visits to the hospital despite having agreed to participate in the trial (Table 2, A4 and A5).

Other, more generic reasons included not wishing to be part of a trial or to receive the placebo, or the presence of other health issues.

In the January–March 2020 period of recruitment, seven in ten patient refusals were due to the extra hospital visits, and although not formally recorded, these may also have been linked to the imminent COVID-19 pandemic.

#### 3.1.2. Enablers to Recruitment

Despite the potential barriers discussed above, several factors identified by our interviewees acted to outweigh them. These included the possibility that the MAB study therapy may enable participants to continue supporting their home life whilst undergoing chemotherapy (Table 2, A6).

The idea of mistletoe as a natural product was also viewed favourably by some, as was the accepted use of this therapy in other countries. Several participants also indicated they had altruistic reasons for taking part (Table 2, A7 and A8). Some participants spoke of the appeal of regaining a sense of self-empowerment that they felt had been lost to cancer, surgery, chemotherapy, and radiotherapy (Table 2, A9).

Staff interviewed also spoke of the enthusiasm they noted for the study in participants, one commenting that some had asked whether they could take part despite not having been approached about the study. The oncologists also noted the positivity generated amongst participants by the trial (Table 2, A10 and A11).

### 3.2. Retention and Adherence

Fourteen participants were randomised and allocated to their MAB study therapy (Figure 2). Two of the fourteen participants withdrew during the trial. Both remained in the study for around five of the anticipated six months’ duration. The first participant withdrew after 4 months and 27 days, and the nurse recorded, “Participant reported ‘forgot’ to take/give injections for approximately 1 month and described feeling no better/no worse so made the decision to stop the treatment.” After unblinding, it was revealed that this participant was on the placebo.

The second participant withdrew at 5 months and 5 days, and the nurse reported, “Patient is currently emotionally distressed and unable to cope with the extra injections [due to COVID-19 outbreak].” After unblinding, it was revealed that this participant was on Iscador M.

Adherence was measured by analysis of patient diaries and questionnaires. Participants were asked at the eligibility stage whether they wanted to complete these online or on paper; 4 of 14 participants opted for online.

The diary data shows that six participants took the MAB study therapy for ≥26 weeks and five participants for ≥20 weeks. The remaining three participants returned diaries for 9, 12, and 16 weeks. In total, 50% of the participants had one break in the MAB study therapy, and two participants had two breaks. All participants who had breaks were on MT, and as per the MAB protocol, which stated that a skin reaction of ≥5 cm should result in a one-week break from the MAB study therapy, these participants had a step-down to a lower dose (Appendix A).

The free-text section in the diaries allowed participants to express how they were coping with the MAB study therapy on a weekly basis. Comments around this were focused on issues with methods of self-administration, pain and discomfort upon administration, difficulty in maintaining the routine of injections, and the impact of the COVID-19 pandemic. There were no comments around the increasing colour and viscosity of the solutions for the active MT as dosages increased, nor any comments on a lack of change in appearance as dosages “increased” in the placebo.

All 14 participants completed the baseline (T0) questionnaire. Twelve questionnaires were returned after the third chemotherapy cycle (T1), and twelve after the last MAB study therapy treatment (T2). The loss of two T1 questionnaires was related to one of the participants who withdrew, and for the other questionnaire, it was unclear why it was not received. At T2, the missing questionnaires were both due to the participants who withdrew.

#### 3.2.1. Barriers to Retention and Adherence

The factors indicated above by the participants who withdrew were also apparent in the interviews, along with other barriers identified by participants who remained in the study: self-injection remained a challenge to the participants.

An additional concern related to the injections was that the syringes were not pre-loaded. Most participants were required to self-inject Filgrastim during their chemotherapy to stimulate white blood cell production. This was provided in a pre-prepared format, and some participants compared their MAB study therapy unfavourably. (Table 2, B1 and B2).

The absence of a visible skin reaction to the MAB study therapy was also a potential barrier to retention and adherence. A research nurse had anticipated that participants randomised to the placebo treatment might withdraw for this reason (Table 2, B3). Practical considerations also played a part in participants’ thinking. The additional hospital visits were an issue, particularly with a lack of available parking nearby (Table 2, B4). The emerging COVID-19 pandemic also impacted and caused additional stress for the participants (Table 2, B5).

For those participants whose treatment regimens included both chemotherapy and radiotherapy, the duration of their MAB study therapy was also a potential barrier to carrying on with it. In some cases, adherence to the MAB study therapy regimen decreased somewhat after completion of chemotherapy. Some interviewees explained that, as they had understood that MT was helpful with chemotherapy side effects, it could seem less important either during radiotherapy or during the additional month of MAB study therapy required post-chemotherapy/radiotherapy (Table 2, B6). Another interviewee offered an additional perspective. When she was able to ring the bell signalling completion of her final radiotherapy session, her MAB study therapy, which was to continue for a further month, became less important (Table 2, B7).

#### 3.2.2. Enablers to Retention and Adherence

The skin reaction around the injection site to the MAB study therapy, although not pleasant, made participants feel they could be receiving active MT (Table 2, B8).

Engagement from family and friends was a further enabler for some participants, and one participant wrote a blog about mistletoe with very positive feedback (Table 2, B9). Personal investment in the trial by participants was also identified by the staff (Table 2, B10). Some interviewees found their own unique ways of remembering their MAB study therapy, such as using a mistletoe-themed song for their morning alarm (Table 2, B11).

The response of the participants to the MAB trial also influenced staff perspectives, some having their reservations over-ridden upon seeing that participants were accepting the therapy and happy with it (Table 2, B12 and B13).

### 3.3. Assessment of Blinding

Ten participants were on an active MT (Iscador M (*n* = 5); Iscador P (*n* = 5)), and four participants were on the placebo treatment. In the final questionnaire (T2), the 12 remaining participants were asked for their thoughts regarding which intervention they thought they had received and were given the options “don’t know”, “active”, or “placebo”. They were also asked to briefly explain their reasoning (Appendix A).

Of the twelve participants, nine were on active mistletoe treatment, and three were allocated to the placebo. Of the 12 responses to the final question around blinding, 6 guessed correctly, 5 did not know, and 1 was incorrect. Those who wrote supporting text (11 responses) made their decision around skin reactions and how they had coped with the treatment and their lives generally. No one described being confident in their decision, and often they described good and bad times. Two participants described the impact of the COVID-19 lockdown on how they felt during cancer treatment. Overall, we cannot be confident of blinding, and future trials will need to continue to address this. Among the ten participants interviewed, it was clear that most had spent considerable amounts of time thinking about whether they were receiving active or placebo therapy whilst on the trial, and their opinions could fluctuate. As with the questionnaire responses, interviewees indicated that the presence or absence of skin reactions and/or their response to chemotherapy influenced their thoughts around this (Table 2, C1–C3).

Some interviewees, though, said they were unable to decide which study therapy they had received as they did not know how their response to chemotherapy would have been in either case (Table 2, C4). Whatever their thoughts on this, however, the research nurses noted that participants were keen to discuss their possible therapy allocation throughout their treatment (Table 2, C5).

One of the nurses indicated they could make a visual distinction between some MT doses and the placebo (Table 2, C6). Whilst this does have implications for the double-blinding of mistletoe therapy trials, as per the protocol, the initial dose was administered by the nurse, who then taught the participant or relative to inject the same dose over the first week. The lower doses of mistletoe therapy are colourless, but if participants asked for help with injections whilst on the higher doses, which do have colour, this could potentially unblind the health professional. We do not have data on whether participants asked for this help or not.

However, while the nurses were able to see both the placebo and MT ampoules, the participants saw only their own trial ampoules. It is of note that no participants on active MT commented on the increasing colour and viscosity of the solutions as dosages increased in the final question around blinding in the T2 questionnaire, the qualitative interviews, or the diaries.

### 3.4. Adverse Events

All adverse events (AEs) experienced by the participants during the MAB trial were collected. In total, 23/218 were related to MAB study therapy injections and experienced by 11 of the 14 participants (Appendix A). Two of the three remaining participants who did not have AEs related to injections were allocated to the placebo group. In total, 13/23 AEs were related to skin reactions to the injections, and as expected, were experienced by the participants receiving MT.

In total, 18 of the 23 AEs were defined as definitely related to the MAB study therapy, with 1 probably, 1 possibly, and 3 unrelated. All 23 AEs were linked to the injections and the skin response to these and were identified as follows: “pain/discomfort with injection” (*n* = 7); “bruising/hematoma at injection site” (*n* = 3); skin reaction, “redness/itchiness, rash/induration/lumps” (*n* = 13).

All AEs were given a CTCAE score of 1 and were described as “not serious” and “expected”. Reporting of intensity was not consistently made, but when reported, it was described as “mild”. The exception was one AE report with a CTCAE of 5. This is considered a reporting error due to other information describing the AE as “not serious” and “expected”. Cross-referencing with the concomitant medications log and the qualitative interviews, this participant was on blood-thinning agents and was vulnerable to excess bruising. Hence, causality is reported as “not related”.

The free-text section in the participant diaries included reflections on these AEs as indicated in the “Retention and Adherence” section above, with one of the main themes focusing on pain and discomfort upon administration. The participant quotes also describe the impacts of these AEs and, as already mentioned, could be viewed as either barriers or as facilitators to retention and adherence by different interviewees (Table 2, E).

### 3.5. Quality-of-Life Questionnaire

Five of the six questionnaires were used to assess quality of life and signs and symptoms associated with cancer treatment. Item completion was high, with the only exception being questions 13 and 14 in the EORTC BR23 and questions 15 and 16 in the FACT-N, which focused on the individual’s sex life or intimate relationships.

Participants often did not respond (completely) to these, particularly in the T1 and T2 questionnaires, and in one case, a participant described a relationship breakdown in an annotation on the form. These questions were also identified as “odd” or “interesting” by some participants in the qualitative interviews (Table 2, E1 and 2).

### 3.6. Outcome Data

This study was not powered to prove the efficacy of MT on quality of life or symptoms of cancer treatment, and no differences were seen within or between groups for these measures.

### 3.7. Complementary and Alternative Medicine Beliefs

The sixth questionnaire was the CAMBI questionnaire, in which participants were asked to rate their opinions on a wide range of statements broadly around complementary and alternative approaches to health using a 0–7 scale (“completely disagree” to “completely agree”) (Appendix A). The results of this questionnaire at TO suggest that, overall, participants were receptive to interventions that allowed them to be involved in their own treatment [items 7, 8, 10] using a natural approach [item 3] that may promote immune boosting [item 4] and self-healing [items 5, 6, 16].

This was reflected in the qualitative interviews. Several participants had a positive view of complementary and alternative medicines (CAMs) and had previously used them; however, others did not have a (strong) opinion on CAM approaches (Table 2, F1 and 2).

Staff interviewees spoke of the possible benefits of complementary therapies for patients, although they indicated they themselves had not needed to use CAM. Although the perspectives of the oncologists interviewed were a little more circumspect, they also saw possible benefits in the use of CAM. (Table 2, F3 and 4)

## 4. Discussion

### 4.1. Summary of Findings

The aim of the MAB study was to test the feasibility of a pilot placebo-controlled, double-blind RCT of MT in patients with breast cancer in the UK NHS setting.

The recruitment rate during the study was ~38% of people approached. Whilst the trial was open to both female and male participants, our recruitment log indicated that only women were approached; this is not unsurprising, as we know that breast cancer is rare in men, with around 400 new cases diagnosed each year in the UK [20].

Most barriers to recruitment for potential participants were the extra treatment and time, unwelcome extra injections, and (to a lesser extent) being allocated to the placebo. It is important to note that this decision was made in the context of physical and emotional challenges of diagnosis, the need for chemotherapy, and the side effects of chemotherapy, including intravenous treatments, use of granulocyte–colony-stimulating factor (GCSF) injections, and other supportive medications. There was a general lack of knowledge about MT expressed by both participants and health professionals.

Once recruited, over three-quarters (12/15) of the participants completed the full trial therapy. Adherence to the therapy regimens was very good, as measured by participant diaries, and so too was the completion of the questionnaires at all three time points. Half of the participants said they did not know their therapy allocation. Both diary data and the qualitative interviews indicated that participants struggled with injections, and in some cases, also with the skin reactions. However, the qualitative interviews show a positive and enthusiastic attitude towards the MAB study therapy, with participants devising ways of coping with the negative aspects of regular self-injection. The number of adverse events related to MAB study therapy was low, and all were related to MAB study therapy injections or skin reactions.

### 4.2. Strengths and Limitations

This was the first clinical trial of MT to be conducted in the UK and in the NHS setting. It shows that MT can be provided and supported through the UK system of care. It also shows that participant adherence to the therapy (placebo or active) was good, as was completion of validated quality-of-life scales.

Whilst there were difficulties in recruiting to the trial, these were not all specific to the MAB study therapy. Our recruitment was a third of that anticipated (15/45). Once optimal recruitment had been established in 2019, the remaining recruitment time was impacted by the impending COVID-19 situation and the ultimate global lockdown in 2020 [21]. In addition to shortening the recruitment time, we feel the COVID-19 situation, with national messages discouraging hospital attendance, as well as fears about infection risk and disruption of some chemotherapy treatment, impacted participation in early 2020.

It was our original thinking to interview people who declined to be involved. Unfortunately, this was not possible, as it was not the researcher who recruited into the trial but rather the research nurse. Once the potential participant expressed that they were not interested in being involved, it did not seem fair to ask them to give their details in order to be contacted by the researcher, and we suspect that our take-up rate would have been low.

The trial was designed by primary and secondary care researchers, patients, and health professionals, but on a day-to-day basis, it was run by secondary care health professionals in an oncology research unit. Despite good communication between the University of Bristol (principal investigators and sponsor) and the Bristol Haematology and Oncology Centre, this resulted in a lag in eliminating glitches in the trial start-up and in some practical aspects of data collection, which were further inhibited by the COVID-19 restrictions. For example, some participants changed diary records from online to paper without formal notification.

Whilst a significant number of potential participants declined involvement due to the extra hospital visits and injections, most who accepted were retained. Both good initial patient information and good participant support from health professionals during the trial will have contributed to this, but we also acknowledge that other factors might have been involved, including, for example, extra attention and a complementary therapy being provided for free on the NHS. We know from both the diaries and the qualitative interviews that participants struggled with the process of regular injections and the skin reactions, yet they persevered with the therapy regimen. It is possible that participants who perceived that they were allocated to the placebo were less motivated.

The qualitative interviews suggest that some participants were less focused on their MAB study therapy after their chemotherapy was complete. The patient information sheet stated clearly that MT could be beneficial during both chemotherapy and radiotherapy. There are likely to be different reasons for wanting to stop the regular self-injections, but the qualitative interviews suggest some participants had just had enough of the injections and wanted to move on.

The lack of awareness of MT and its potential responses likely enhanced the blinding of participants to their therapy allocation. The increasing colour and viscosity of the solutions for the active mistletoe treatment as dosages increased was not commented on by the participants in the qualitative interviews, diaries, or the final question around blinding. The appearance of the saline placebo would remain identical throughout, but due to the lack of knowledge around MT, this equally may not have been a factor for participants.

Skin reactions and especially prolonged skin reactions are much less likely with the saline placebo. From the diary data, two of the four participants on placebo had at least one episode of minor skin reactions, and one participant also had a skin infection, which may have been misinterpreted as a reaction.

While health professionals were not formally asked for their perspective on the allocated treatments, the qualitative interviews suggested that one nurse was aware of the increasing colouration of some participants’ therapies. We acknowledge that there is a potential for unblinding with higher doses of mistletoe. The number of adverse events was low, entirely related to self-injections and the skin reactions expected with MT. This is in line with adverse event monitoring in previous clinical trials [22,23].

We used a range of quality-of-life measures and the CAMBI questionnaire. We can provide access to the data with no comparative analysis, as appropriate for a feasibility study. Equally, we cannot comment on the comparative effectiveness of the two Iscador products beyond the fact that having two active therapies may have enhanced recruitment. Although overall the quality-of-life questionnaires were completed well, the questions relating to sexual feelings, sex life, and interpersonal relationships in EORTC BR23 and FACT-N were often not completed, and two interviewees’ comments suggested that these items were less relevant to them during breast cancer treatment. Further exploration of the inclusion of these items is warranted.

### 4.3. Implications for Clinical Practice and Further Research

MT is integrated into oncology care via health insurance schemes in Germany and Switzerland. In the UK, it is predominantly available via private practice, although there is some NHS provision [5]. An HTA report that focused on adjunct MT for breast cancer patients concluded that improvements in health-related quality of life compared to the control group were small to moderate, but that the risk of bias within these efficacy trials is high [4]. Broader reviews across cancer types come to a similar conclusion [24,25,26].

Our study design did not permit us to determine a primary endpoint, effect size, or sample size. However, published full RCTs with a similar population conducted in other countries and settings suggest the use of EORTC-QLQ-C30 is preferable for quality-of-life measurement, and the two Semiglasnov RCTs of breast cancer patients provide estimates of effect size and calculate a sample size [11,27,28,29].

Whilst it is unlikely that MT will be made more widely available on the NHS in the near future, MT is used by cancer patients in the UK. MT is available across anthroposophic medicine centres in England and Scotland, and is likely to be available through some individual General Practitioners. It is, therefore, important that we continue to add to the evidence base of this safe adjunct therapy for cancer patients. Previous qualitative research and the interviews within the MAB study suggest a perceived benefit for cancer patients [30,31].

Research based in the UK should focus on mistletoe provision within its countries, adding to the evidence base of efficacy within 21st-century conventional cancer treatment. It is also important that different modes of application of mistletoe are examined, bearing in mind the challenges of self-injection highlighted by this mixed-methods trial. Other potential modes include pre-filled syringes, oral mistletoe, and hydrogels [32,33]. Such modalities would further enhance the acceptability of mistletoe therapy by both patient and practitioner, but to date, there is little to no research on these approaches.

## 5. Conclusions

This pilot study examined the feasibility of conducting a mixed-phase, pilot, randomised, placebo-controlled trial of mistletoe therapy for breast cancer patients within the UK National Health Service. The results describe the challenges and achievements of recruitment, retention, adherence, blinding, and safety in this context.

## Figures and Tables

**Figure 1 cancers-17-03169-f001:**
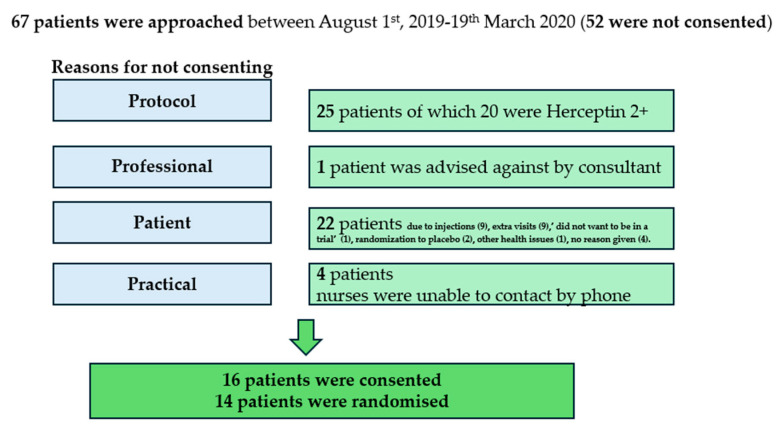
Screening and recruiting for the MAB study.

**Figure 2 cancers-17-03169-f002:**
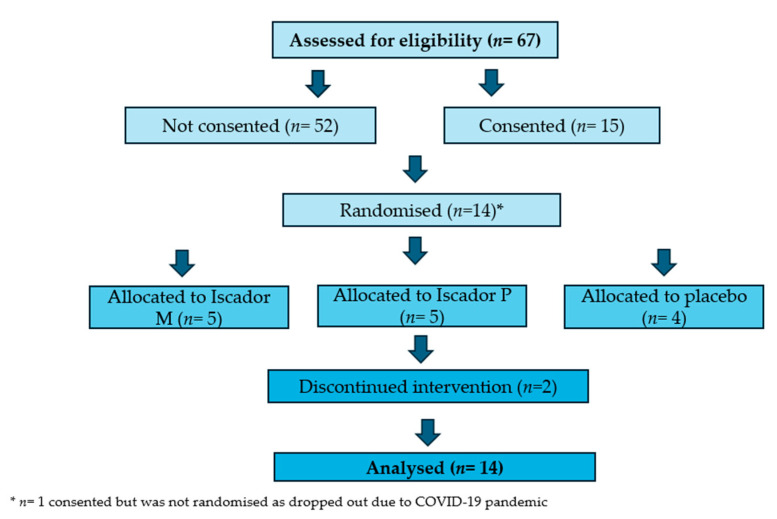
Consort diagram.

**Table 1 cancers-17-03169-t001:** Characteristics of MAB participants (n = 14).

Mean Age (yrs)	Age Range (yrs)	Identified As	Education	Occupation	Laterality	Stage	Tumour Size	Nodes	ER Status
49	36–76	White British = 11 White other background = 3 Chinese = 1	No formal qualifications *n* = 1 Secondary *n* = 2 Further *n* = 2 Higher *n* = 9	Employed *n* = 11 retired *n* = 1 unemployed due to ill health *n* = 1	Right = 7 Left = 7	Stage1 = 3 Stage2 = 9 Stage3 = 2	T1 = 2 T2 = 12	N0 = 7 N1 = 4 N2 = 2 N3 = 1	ER + = 10 ER − = 4

Key: Secondary education: ≤16 yrs (level 2). Further education: 16–19 yrs (level 3). Higher education: 18 yrs + (Level 4+).

**Table 2 cancers-17-03169-t002:** Illustrative quotes from participant interviews.

Feasibility Outcomes	Illustrative Quotes
**A Recruitment**	
Awareness of mistletoe therapy	**A1** “Never heard of it before.” Participant G. **A2** “The berries are poisonous, so this was quite interesting. I was like ‘mistletoe?’ (laughs) Gosh, ok.” Participant K. **A3** “A while back it was quite a popular thing for people to try … for their metastatic cancer. A lot of patients were looking at getting that in the private sector.” Oncologist A.
Barriers to recruitment	**A4** “If it will help me then I’ll have it, but I find it hard to inject myself.” Participant G. **A5** “Initially I said no ‘cos of all the extra hospital appointments … with the little one and we live out at (village X) so getting to the hospital (is difficult).” Participant B.
Enablers to recruitment	**A6** “I thought it was a chance to maintain my family life by reducing my side effects.” Participant H. **A7** “Having a natural product that could alleviate something like a chemo treatment, I found the idea of that absolutely amazing.” Participant J. **A8** “I said ‘yes I’d like to go ahead with it’ because I did see that Germany, Switzerland and Holland had already started using it and I did a bit of research and people were paying for it privately in America and this country …. I just thought anything that will give me an edge as well is absolutely going to help. Help me and also other people for the future, so it’s win-win.” Participant J. **A9** “If I am finding this too much, I can say no… the other things I didn’t have control over.” Participant C. **A10** “[Patients] question if there is any trial they would be eligible for, having mistletoe, because I think they in a way are aware …. The thing is patients are interested; I find them to be more interested.” Research Nurse A. **A11** “[Patients were] clearly enthusiastic about potentially entering the study …. Happy to do any extra attendances that might be necessary.” Oncologist A.
**B Retention and adherence**	
Barriers to retention and adherence	**B1** “When you don’t like needles then you’ll always be terrified by needles.” Participant H. **B2** “You have to build the injection up yourself … in some ways I found that quite difficult because you’ve got time to think about what you’re doing and I’d rather not, I’d rather just take something out of a packet and off. Yeah. So in a way it prolonged the agony.” Participant J. **B3** “I kind of feel like if someone notices that she’s not reacting then she probably will think she’s on placebo and may drop off. So far it didn’t happen but I can’t really say it’s not going to.” Research Nurse A. **B4** “It’s proven to be a bit tiresome, I must be honest with you, having to come into oncology … we have to keep going back weekly and it means coming into the city and it’s a nightmare to park and all of those things.” Participant D. **B5** “Well I have to say I didn’t like going [to the hospital]… it did make it more stressful because obviously the place you don’t want to go to is the place you’ve got to go to, but they temperature checked you at the door on the way in, I had a mask, I had gloves, they had masks and you just had to get on with it really.” Participant F. **B6** “I had the first week of radio and then ... I kept forgetting to take it.” Participant A. **B7** “Once I’d finished the other treatment then the mistletoe became kind of secondary …. After my last radiotherapy I rang the bell and my husband and my daughter were there with me and that was a real kind of emotional moment and it felt like a real sense of closure and yet with the mistletoe, that was still going on, so it wasn’t a closure.” Participant E.
Enablers to retention and adherence	**B8** “It was painful and it was irritating me, I was like ‘oh I can’t be doing with this’ and I did have thoughts ‘oh shall I just finish with it?’ And then in the back of my mind I just thought ‘well would I really be having these reactions if I was on placebo? I just kind of had to ride the storm basically and I’m glad I did because I ended up sort of talking to myself going right, ok, there are positives to this, this is uncomfortable at the moment but, you know, it’s not going to last.’” Participant C. **B9** “I had a blog page …. I told them all about the mistletoe on that and I got a very positive response.” Participant D. **B10** “Even if they’d said they got a skin reaction and we’d said ‘ok, tell us about that, is that problematic’, they were still really keen to kind of carry on and say ‘no, I really want to do this and it’s fine’ …. So once they’d signed up to it they really wanted to persevere and see it through.” Research Nurse C. **B11** “I’ve got an alarm on my phone that goes off in the morning, I’ve chosen Justin Bieber’s classic hit ‘Mistletoe’ to remind me!” Participant E. **B12** “What we were concerned about is the injecting themselves. So they seem to be coping alright … and they all seem to be sort of happy to be carrying on with it which is actually more surprising.” Research Nurse B. **B13** “To begin with I was a bit uncertain, but when I saw they were accepting it quite easily I thought that was quite good and they didn’t have any side effects. And none of them complained to me about being tired, whereas 99% of the people, if you look at any of my letters …. would have toxicity and fatigue.” Oncologist B.
**C Assessment of blinding**	**C1** “What makes me think I might have had the placebo because I’ve had no skin reactions at all.” Participant F. **C2** “I viewed [the skin reaction] as possibly having the actual mistletoe injection instead of a placebo so I was like this is good, I’ve actually got the mistletoe (laughs) so I was quite pleased ... I think with the reaction that I had and the way that I felt throughout chemotherapy, I would lean more towards thinking that I had had either mistletoes, but like I say you never know, maybe it was all in my mind. But I think, yeah, I think that I did have mistletoe.” Participant B. **C3** “I feel very strongly that it was the mistletoe …. I was ready for the [chemotherapy] side effects and I didn’t have them … All I had was hunger.” Participant K. **C4** “I don’t know if it was the mistletoe that sort of helped or if it was just the frame of mind, I honestly couldn’t tell you.” Participant A. **C5** “Some people were saying I don’t think I had it because I still had some side effects or there were others that said I think I did have it because I felt great all the way through, but I think they definitely had an opinion as to whether they were on it.” Research Nurse C. **C6** “So the colour of the [mistletoe] liquid is probably a light yellow but the saline is … colourless.… I can say from week four I can see the difference.” Research Nurse A.
**D Adverse events**	Relevant quotes in B and C, e.g., **B8, B9, and C2**
**E Questionnaire data**	**E1** “There was some slightly odder questions than others. There was a couple of questions about my sex life which I thought was interesting, a bit left-field.” Participant E. **E2** “... there were questions about kind of, you know, like your sex life and that sort of… Some of that obviously, yeah, ... I just put not applicable.” Participant C.
**F Complementary and alternative medicine beliefs**	**F1** “I fully understand the power of plants ... But I don’t fully believe they can cure everything ‘cos I think there is a place for engineered drugs if you need them. I think it’s a complementary thing.” Participant F. **F2** “To be perfectly honest I don’t really think about it … just go with the flow.” Participant A, who had not used CAM therapies previously. **F3** “I think mainstream treatment should absolutely sit alongside kind of complementary or additional therapies because if it’s improving their general health, wellbeing, emotional and physical kind of health then that’s great.” Research Nurse C. **F4** “I tend to be fairly relaxed about patients taking …. complementary medicines because I think at the very least you’ll be harnessing a placebo and potential psychological benefits, and then there may be added benefits. Now if it’s got a very active ingredient or there’s something unknown about it then I’d be more cautious ‘cos I think we just don’t have any evidence about the interactions.” Oncologist A.

## Data Availability

Any further data not presented within the paper or supporting documents are available from the authors upon reasonable request.

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
