# Peer review of "The Mistletoe and Breast Cancer (MAB) Study: A UK Mixed-Phase, Pilot, Placebo-Controlled, Double-Blind, Randomised Controlled Trial"

_cancers, 2025, doi:10.3390/cancers17193169_

Round 1

Reviewer 1 Report

Comments and Suggestions for Authors

Overall I think this is an important piece of work and a well presented pilot study. The authors have overcome many obstacles on the way and succeeded in bringing this pilot trial to a closure, which should be commended. It is also plausible with a mixed-methods design, although this comes with some problems too. Although my overall assessment calls for major revisions I do hope this will result in a final publication of this paper. Please find my comments below.

2.8.4. Qualitative interviews

Please describe the role of the interviewer LD in relation to the study participants. Also, describe any preunderstanding of LD that may influence the interviews.

Those interviews conducted in person, please tell where were they conducted since place of interview may influence outcomes greatly.

2.9. Data analysis

Please also provide information about the theoretical framework and framework for analysis that was used for analysing the qualitative data.

Results: Row 240-247

Table 2 provides a clear description of the findings but please add a summary of the findings in the text before reporting specific results.

Row 267

“These concerns were echoed in the qualitative interviews. [Table 2,A4 & A5]”

Please clarify your interpretation of these citations.

Figure 1 has a good intention but needs clarification.

Row 386-387

Please clarify the pracitioners’ comments about the increased color in the ampules (did more than one nurse make such a comment?). This is important since it is potentially a threat to the double blind design.

3.6 Outcome data

Since the sample is not big enough to detect differences and this is a feasability study I question the reason for including a presentation of the global scores of EORTC. I would recommend erasing this presentation, saving this for the larger sample.

Discussion

Row 502-503: “most who accepted were retained. This reflects both good initial patient information and good participant support from health professionals during the trial. “ I think this interpretation needs to be more cautions. There are various other possible reasons for this, one of them being the offer of a complementary therapy for free within NHS serviec.

Row 526-527

“…one nurse was aware of the 526 increasing colouration of some participants’ therapies.” Please elaborate a little more on this since it is essential for the double blind design.

Row 534-535

“…Of note, although overall the quality-of-life questionnaires were completed well, 534 the questions relating to sexual feelings, sex life and interpersonal relationships appear to 535 be inappropriate for our population at this point in their treatment.” How do you know that the questions were percieved as “inappropriate”? I would elaborate more carefully on this if there is no more information from the interviews about this.

Please look into if there is anything about this in the interviews. If there is a lack of answers here, this is an important pointer for exploration in the interviews coupled to the future trial. This issue deserves discussion in relation to the literature that is abound about this group of patients and their experiences.

Author Response

Reviewer one

Overall, I think this is an important piece of work and a well presented pilot study. The authors have overcome many obstacles on the way and succeeded in bringing this pilot trial to a closure, which should be commended. It is also plausible with a mixed-methods design, although this comes with some problems too. Although my overall assessment calls for major revisions I do hope this will result in a final publication of this paper. Please find my comments below.

2.8.4. Qualitative interviews

Please describe the role of the interviewer LD in relation to the study participants. Also, describe any preunderstanding of LD that may influence the interviews.

We have added this text:

LD is an experienced qualitative researcher based at the University of Bristol with a research interest in complementary therapies. Participating patients were initially approached by the research nurses, if they agreed, the research nurses passed their contact details to LD who then sought consent and arranged an interview. LD approached BHOC staff directly. L171-175.

Those interviews conducted in person, please tell where were they conducted since place of interview may influence outcomes greatly.

These five interviews were conducted in a place agreed with the participants: Canynge Hall (University of Bristol) (2), quiet spots within public cafes in Bristol (2)and  in a participants work place (1). L179-181

2.9. Data analysis

Please also provide information about the theoretical framework and framework for analysis that was used for analysing the qualitative data.

We have amended and added this text:

Interviews were transcribed and coded in NVivo [version 12.0] using themes broadly linked to those used in the interview topic guides. [Appendices D & E]  A qualitative inductive analysis approach was used, guided by Braun and Clarke methods [14] Initial coding and development of themes were performed by LD and AH, with the remaining coding and synthesis performed by LD, MG and PG. Narrative synthesis of the themes from the perspectives of participants and staff are reported using the interview data as well as relevant items from participant diaries and questionnaires [15]. L206-212.

Results: Row 240-247 Table 2 provides a clear description of the findings but please add a summary of the findings in the text before reporting specific results.

We have added this text:

Qualitative analysis of the interviews was based on the feasibility outcome framework A-F with illustrative quotes. [Table 2]. A) Recruitment:  awareness of mistletoe therapy, barriers to recruitment, enablers to recruitment. B) Retention and adherence: barriers to retention and adherence, enablers to retention and adherence C) Assessment of blinding D) Adverse Events, E) Questionnaire data and F) Complementary and alternative medicine beliefs. L249-254

Row 267 “These concerns were echoed in the qualitative interviews. [Table 2,A4 & A5]”Please clarify your interpretation of these citations.

We have added this text:

With some participants expressing reservations around self-injections and extra visits to the hospital despite having agreed to participate in the trial. L282-284

Figure 1 has a good intention but needs clarification.

We have redesigned Figure 1 to give more detail and clarity  L290

Row 386-387 Please clarify the practitioners’ comments about the increased colour in the ampules (did more than one nurse make such a comment?). This is important since it is potentially a threat to the double-blind design.

Only one nurse noted this.

We have added this text:

Whilst this does have implications for double-blinding of mistletoe therapy trials, as per the protocol the initial dose was administered by the nurse who then taught the participant or relative to inject the same dose over the first week. The lower doses of mistletoe therapy are colourless but if participants asked for help with injections whilst on the higher doses which do have colour, this could potentially unblind the health professional. We do not have data on whether participants asked for this help or not. L404-410

3.6 Outcome data

Since the sample is not big enough to detect differences and this is a feasibility study I question the reason for including a presentation of the global scores of EORTC. I would recommend erasing this presentation, saving this for the larger sample.

Agree. We have removed this.

Discussion

Row 502-503: “most who accepted were retained. This reflects both good initial patient information and good participant support from health professionals during the trial. “ I think this interpretation needs to be more cautions. There are various other possible reasons for this, one of them being the offer of a complementary therapy for free within NHS service.

We have edited and added to this statement:

Both good initial patient information and good participant support from health professionals during the trial will have contributed towards this, but we also acknowledge that other factors might have been involved including for example extra attention and a complementary therapy being provided for free on the NHS.L529-532

Row 526-527“…one nurse was aware of the 526 increasing colouration of some participants’ therapies.” Please elaborate a little more on this since it is essential for the double blind design.

We have added

‘We acknowledge that there is a potential for unblinding with higher doses of mistletoe.’ L556-557.

Which complements the text below which has already been added into results.

Whilst this does have implications for double-blinding of mistletoe therapy trials, as per the protocol the initial dose was administered by the nurse who then taught the participant or relative to inject the same dose over the first week. The lower doses of mistletoe therapy are colourless but if participants asked for help with injections whilst on the higher doses which do have colour, this could unblind the health professional. We do not have data on whether participants asked for this help or not.

Row 534-535 “…Of note, although overall the quality-of-life questionnaires were completed well, 534 the questions relating to sexual feelings, sex life and interpersonal relationships appear to 535 be inappropriate for our population at this point in their treatment.” How do you know that the questions were perceived as “inappropriate”? I would elaborate more carefully on this if there is no more information from the interviews about this. Please look into if there is anything about this in the interviews. If there is a lack of answers here, this is an important pointer for exploration in the interviews coupled to the future trial. This issue deserves discussion in relation to the literature that is abound about this group of patients and their experiences.

We have edited this to say:

Of note, although overall the quality-of-life questionnaires were completed well,  the questions relating to sexual feelings, sex life and interpersonal relationships in EORTC BR23 and FACT-N were often not completed, and two interviewees comments suggested that these items were less relevant to them during breast cancer treatment. Further exploration into the inclusion of these items is warranted.L564-568

Reviewer 2 Report

Comments and Suggestions for Authors

The authors provided a detailed description of randomized controlled study for mistletoe therapy. Unlike other drugs used for cancer treatment, mistletoe has been used since prehistoric times and lacks a well-designed clinical trial to test its efficacy. Since immune checkpoint inhibitors are being used for breast cancer, mistletoe for cancer treatment should be investigated within the frame of evidence-based medicine.

The study provides valuable information for clinical researchers who perform randomized controlled trials and may serve as a foundation for future studies. However, it only included 14 patients and did not provide a solid conclusion. Presenting the feasibility of randomized controlled trials does not guarantee publication as an original article. I recommend the authors publish the study as a research protocol or brief communication.

Some specific points:

  • Line 54: What does “HTA” mean? There is no description for this abbreviation.
  • Line 109, 112, 114: The article uses “MT (mistletoe therapy)” and “MAB (mistletoe and breast cancer) therapy”. Are there any differences between the two terms? If not, please unify the terms.
  • Line 201-202, 213: The initials of each team member are unnecessarily described in the text.
  • Table 2: A detailed description of interview can be provided as a supplement.

Author Response

Reviewer two

The authors provided a detailed description of randomized controlled study for mistletoe therapy. Unlike other drugs used for cancer treatment, mistletoe has been used since prehistoric times and lacks a well-designed clinical trial to test its efficacy. Since immune checkpoint inhibitors are being used for breast cancer, mistletoe for cancer treatment should be investigated within the frame of evidence-based medicine.

The study provides valuable information for clinical researchers who perform randomized controlled trials and may serve as a foundation for future studies. However, it only included 14 patients and did not provide a solid conclusion.

Presenting the feasibility of randomized controlled trials does not guarantee publication as an original article. I recommend the authors publish the study as a research protocol or brief communication.

Response

We agree with the reviewer that the MAB study does not provide a solid conclusion on efficacy of mistletoe therapy, but it does provide valuable feasibility data outlined in our study aims. The protocol is already published, the study was registered as a clinical study, randomised and designed to be double-blind and has provided much important information/design ideas in conducting a full scale trial in the UK.  Neither the protocol nor the final paper makes any claims around efficacy and thus its title includes the term pilot.

We have edited the title of the paper to fully align it with the published protocol:

The Mistletoe And Breast cancer (MAB) study: A UK mixed-phase, pilot, placebo-controlled, double-blind, randomised controlled trial. L2-4

Some specific points:

Line 54: What does “HTA” mean? There is no description for this abbreviation.

The full term, Health Technology Assessment and its abbreviation, has been added to the first sentence of the second paragraph of the introduction. L46

Line 109, 112, 114: The article uses “MT (mistletoe therapy)” and “MAB (mistletoe and breast cancer) therapy”. Are there any differences between the two terms? If not, please unify the terms.

The title of the study is MAB; the name of the therapy is mistletoe therapy (MT). We have used the term MAB therapy to include both MT and placebo treatment – we agree this is confusing

We have edited all mentions of MAB therapy to say MAB study therapy  as required throughout the document.

Line 201-202, 213: The initials of each team member are unnecessarily described in the text.

These have been removed.L224-236

Table 2: A detailed description of interview can be provided as a supplement.

Topic guides are available in appendices D & E.  More detail of the process is in the published protocol paper.

Reviewer 3 Report

Comments and Suggestions for Authors

The present work aims to test the feasibility of a pilot, randomised controlled trial of mistletoe therapy (MT) with an embedded qualitative study in the UK National Health Service (NHS) setting. Although the study could of interest, there are several flaws: 

Title – the title is misleading, as it gives the impression of a completed placebo-controlled RCT. In fact, the purpose of the manuscript, as presented, was only to test the FEASIBILITY of a pilot, mixed phase, mixed methods, and placebo-controlled RCT. I would recommend rephrasing the title to emphasise the feasibility nature of the work.

General

When citing, in-text references should be inserted prior to the final punctuation of the sentence.

Abstract

The abstract states that 45 patients were recruited. According to the trial, only 15 patients were recruited, and 14 were randomized. Please revise.

The statement ‘Adherence was very good’ should be linked to the fact that it is based on only 12 completers. Please rephrase to make clear that this refers to the small sample analysed.

"Diaries and interviews indicated that some participants struggled with injections and skin reactions." (line 30) - Please consider rephrasing this statement with the actual numbers (ex 11/14 participants, lines 392-393), as ‘some’ underestimates the proportion of patients affected.

Manuscript

The accuracy of reference [1] is unclear to me. Is this a reliable source?

Line 54 – explain what HTA is

What are the official opinions and recommendations of the NHS, FDA, and EMA regarding the use of mistletoe in cancer care? Please include these, if available.

Clarify randomisation: It is stated that patient randomisation lists were generated by an in-house statistician at Iscador AG, which may present a conflict of interest. Please describe measures taken to ensure impartiality and allocation concealment.

The results (lines 362-390) reported suggest that blinding was only partially successful; many participants correctly identified their allocation. Please acknowledge this limitation more clearly in the discussion.

The clarity of Figure 1 and Figure 2 should be improved.

Lines 440-442 – “The results of this questionnaire at T0 suggest that overall participants were in favour of interventions that allow them to be involved in their own treatment, are considered natural and may promote immune-boosting and self-healing” – The terms ‘natural’ and ‘immune-boosting’ used here do not seem to correspond to specific items listed in Appendix H. Please clarify how these terms were derived from the actual questionnaire responses.

Please revise also in the abstract “Participants perceived MT as natural, immune-boosting and promoting self-healing”.  I could not find evidence in Appendix H that patients explicitly described MT as ‘immune-boosting’ or ‘promoting self-healing’ (although they did mention, for example, that treatments should enable the body to heal itself). Please revise to be more accurate.

Line 454 - the manuscript reports that recruitment was ‘optimised to just under 40%’, but in reality, the recruitment was only 16% for the first 5 months and increased to ~38% only after a major protocol amendment. This should be stated more transparently.

The study initially aimed to recruit 45 participants but only enrolled 15, of whom 12 completed the intervention (or of the 67 individuals approached, only 15 provided informed consent). The authors acknowledge that recruitment fell short due to COVID-19 and restrictions, yet still conclude the trial demonstrated feasibility. Since recruitment is a core feasibility parameter (so is blinding), the conclusion should be toned down to reflect that only some aspects of feasibility were achieved. Also, the general tone of the results would benefit from being more cautious and explicitly grounded in the small sample size on which the findings are based (ex. line 467).

Author Response

Reviewer 3

The present work aims to test the feasibility of a pilot, randomised controlled trial of mistletoe therapy (MT) with an embedded qualitative study in the UK National Health Service (NHS) setting. Although the study could of interest, there are several flaws:

Title – the title is misleading, as it gives the impression of a completed placebo-controlled RCT. In fact, the purpose of the manuscript, as presented, was only to test the FEASIBILITY of a pilot, mixed phase, mixed methods, and placebo-controlled RCT. I would recommend rephrasing the title to emphasise the feasibility nature of the work.

See response to  reviewer 2 (detailed below)

We agree with the reviewer that the MAB study does not provide a solid conclusion on efficacy of mistletoe therapy, but it does provide valuable feasibility data outlined in our study aims. The protocol is already published, the study was registered as a clinical study, randomised and designed to be double-blind and has provided much important information/design ideas in conducting a full scale trial in the UK.  Neither the protocol nor the final paper make any claims around efficacy and thus its title includes the term pilot.

We have edited the title of the paper to fully align it with its published protocol.

The Mistletoe And Breast cancer (MAB) study: A UK mixed-phase, pilot, placebo-controlled, double-blind, randomised controlled trial.

General

When citing, in-text references should be inserted prior to the final punctuation of the sentence.

Corrected throughout manuscript

Abstract

The abstract states that 45 patients were recruited. According to the trial, only 15 patients were recruited, and 14 were randomized. Please revise.

This has been amended to state that this was the original aim. L24

The statement ‘Adherence was very good’ should be linked to the fact that it is based on only 12 completers. Please rephrase to make clear that this refers to the small sample analysed.

Rephrased to:

Adherence was very good whilst the participants were on the study therapy.L32-33

"Diaries and interviews indicated that some participants struggled with injections and skin reactions." (line 30) - Please consider rephrasing this statement with the actual numbers (ex 11/14 participants, lines 392-393), as ‘some’ underestimates the proportion of patients affected.

This has been amended L34

Manuscript

The accuracy of reference [1] is unclear to me. Is this a reliable source?

We have removed the first sentence ‘Mistletoe ( viscum album) has been used medically since prehistoric times.’

Reference one outlines anthroposophical  history/use of mistletoe within anthroposophical medicine.– Whilst only a  website reference it is the most detailed description of such use – with the references cited within it referring to individual studies and systematic reviews.

Line 54 – explain what HTA is

Done

What are the official opinions and recommendations of the NHS, FDA, and EMA regarding the use of mistletoe in cancer care? Please include these, if available.

The following information is given in the introduction

The NHS do not specifically give information or recommendations on MT, but  it does advise caution to cancer patients taking herbal therapy generally during their treatment [https://www.nhs.uk/tests-and-treatments/herbal-medicines/] L73-75

Neither the European Medicines Agency (EMA) nor the Food and Drug Administration (FDA) state any opinions and recommendations on mistletoe therapy (MT) although they have both compiled summary of MT evidence. L47-50

https://www.ema.europa.eu/en/documents/herbal-report/final-assessment-report-viscum-album-l-herba_en.pdf

https://www.cancer.gov/about-cancer/treatment/cam/patient/mistletoe-pdq

Clarify randomisation: It is stated that patient randomisation lists were generated by an in-house statistician at Iscador AG, which may present a conflict of interest. Please describe measures taken to ensure impartiality and allocation concealment.

The is a detailed description of the randomisation process in the protocol paper.  We have cited this paper at this point.

Details of the randomisation process are available in the protocol paper. L118

The results (lines 362-390) reported suggest that blinding was only partially successful; many participants correctly identified their allocation. Please acknowledge this limitation more clearly in the discussion.

Added to the text:

Of the 12 responses to the final question around blinding, 6 guessed correctly, 5 did not know and one was incorrect. Those that wrote supporting text (11 responses)  made their decision around skin reactions and how they had coped with the treatment and their lives generally.  No one described being confident in their decision, and often they described good and bad times. Two participants described the impact of the COVID-19 lockdown on how they felt during cancer treatment. Overall we cannot  be confident of blinding and future trials will need to continue to address this. L400-406.

The clarity of Figure 1 and Figure 2 should be improved.

Done  

Lines 440-442 – “The results of this questionnaire at T0 suggest that overall participants were in favour of interventions that allow them to be involved in their own treatment, are considered natural and may promote immune-boosting and self-healing” – The terms ‘natural’ and ‘immune-boosting’ used here do not seem to correspond to specific items listed in Appendix H. Please clarify how these terms were derived from the actual questionnaire responses.

The text has been edited to:

The results of this questionnaire at TO suggest that overall participants were receptive to interventions that allowed them to be involved in their own treatment [items 7,8,10] using a natural approach [item 3] that may promote immune boosting [item 4]  and self-healing [items 5,6,16].L486- 489

Please revise also in the abstract “Participants perceived MT as natural, immune-boosting and promoting self-healing”.  I could not find evidence in Appendix H that patients explicitly described MT as ‘immune-boosting’ or ‘promoting self-healing’ (although they did mention, for example, that treatments should enable the body to heal itself). Please revise to be more accurate.

We have decided to take this sentence out of the abstract  as it is difficult to articulate in one sentence, The word count in the abstract is 150 words and there are more important things to emphasise.

Line 454 - the manuscript reports that recruitment was ‘optimised to just under 40%’, but in reality, the recruitment was only 16% for the first 5 months and increased to ~38% only after a major protocol amendment. This should be stated more transparently.

We have edited this sentence in the discussion to state

The recruitment rate during the study was ~38% of people approached. L504

The study initially aimed to recruit 45 participants but only enrolled 15, of whom 12 completed the intervention (or of the 67 individuals approached, only 15 provided informed consent). The authors acknowledge that recruitment fell short due to COVID-19 and restrictions, yet still conclude the trial demonstrated feasibility.

 Since recruitment is a core feasibility parameter (so is blinding), the conclusion should be toned down to reflect that only some aspects of feasibility were achieved. Also, the general tone of the results would benefit from being more cautious and explicitly grounded in the small sample size on which the findings are based (ex. line 467).

We have edited the concluding sentence in both the abstract and discussion to say

This pilot study examined the feasibility of conducting a pilot randomised controlled double-blind trial of mistletoe therapy of breast cancer patients within the UK National Health Service. The results describe the challenges and achievements of recruitment, retention, adherence, blinding and safety in this context. L38-41, L628-633

Round 2

Reviewer 1 Report

Comments and Suggestions for Authors

I thank the authors for very good and relevant revisions! I think the paper is ready for publication.   

Author Response

Reviewer one has made no further comments on the text of the paper.

Reviewer 2 Report

Comments and Suggestions for Authors

Line 127, 129, 175, 194, 317, 330, 344, 346, 349, 352, 359, 374, 376, 384, 385, 388, 394, 399, I recommend using the term ”mistletoe therapy” instead of “MAB study therapy”. The study protocol is not novel; rather, it follows a standard approach of administering mistletoe.

Line 230-231. It is unnecessary to introduce team members in a scientific article.

Author Response

Reviewer two has made no further comments on the text of the paper.  We had previously improved the figures, we have now improved table 1 and 2 which originally were formatted for landscape.

Reviewer 3 Report

Comments and Suggestions for Authors

The authors have implemented the recommended changes, which have improved the quality of the manuscript. In my opinion, it is now suitable for publication. Congratulations, and best wishes for future projects!

Author Response

Reviewer three has made no further comments on the text of the paper.